# The rise of angiosperms strengthened fire feedbacks and improved the regulation of atmospheric oxygen

Claire M. Belcher[1,2 ✉], Benjamin J. W. Mills [3], Rayanne Vitali[1,2], Sarah J. Baker[1,2], Timothy M. Lenton [2] & Andrew J. Watson [2]

The source of oxygen to Earth's atmosphere is organic carbon burial, whilst the main sink is oxidative weathering of fossil carbon. However, this sink is to insensitive to counteract oxygen rising above its current level of about 21%. Biogeochemical models suggest that wildfires provide an additional regulatory feedback mechanism. However, none have considered how the evolution of different plant groups through time have interacted with this feedback. The Cretaceous Period saw not only super-ambient levels of atmospheric oxygen but also the evolution of the angiosperms, that then rose to dominate Earth's ecosystems. Here we show, using the COPSE biogeochemical model, that angiosperm-driven alteration of fire feedbacks likely lowered atmospheric oxygen levels from ~30% to 25% by the end of the Cretaceous. This likely set the stage for the emergence of closed-canopy angiosperm tropical rainforests that we suggest would not have been possible without angiosperm enhancement of fire feedbacks.

[1] wildFIRE Lab, University of Exeter, Exeter EX4 4PS, UK. [2] Global System Institute, University of Exeter, Exeter EX4 4QE, UK. [3] School of Earth and Environment, University of Leeds, Leeds LS2 9TJ, UK. ✉email: c.belcher@exeter.ac.uk

Fire and vegetation, the fuel for fires, have a two-way interaction with each other, the environment and the atmosphere. For example, many terrestrial ecosystems are considered fire prone, where the biological and chemical traits of plants influence fire regimes but equally over time fire has shaped the composition and structure of fire prone ecosystems[1–5]. Fire regimes are themselves controlled by atmospheric conditions, composition, climate and the type of vegetation[5,6] whilst the effects of fires, the products and emissions they generate influence biogeochemical cycles and long-term Earth system processes through their impacts on nutrient cycles and by altering the composition and distribution of biomes[7–10]. Hence fire is more than a simple agent of disturbance and can generate multiple complex feedbacks.

Wildfires have existed as a physical force that has acted on our ecosystems for some 420 million years[11]. Many of our extant plants show physiological and life-history traits that many ecologists have interpreted as adaptations to fire. An example is that of the serotinous cones of some conifer species that appear to be adapted to opening only in response to intense wildfires, and in doing so shed their seeds to ensure reestablishment of a new generation after fire[12]. This adaptive trait in conifers is believed to have evolved millions of years ago in the Late Cretaceous period, when wildfires were both more frequent[13] and more intense[14]. Major evolutionary events have occurred in the plant world throughout evolutionary time. For example, variations in fire regimes have been linked to the evolution of new leaf properties with the appearance of flowering plants (the angiosperms) and grasses (the Poaceae)[14–16] that altered plant productivity and the ability to survive in open environments. As such plant innovations and fire are intimately linked. Despite this long inter-related multi-million year history of plants and fire the extent to which evolutionary innovations of fuel have altered fire feedbacks to the Earth system have not been well explored.

Fire is extremely responsive to the level of oxygen in the atmosphere. Its presence, as evidenced by the appearance of charcoals in the fossil record, has been suggested to indicate that the abundance of oxygen in the atmosphere must be >16%[17,18]. For almost half the history of our planet there were only trace levels of oxygen in the atmosphere. However, over the past ~2 billion years atmospheric oxygen has risen from trace to present-day levels, most likely in a series of steps, due to the evolution of oxygenic photosynthesis and subsequent shifts in the redox state of the ocean and atmosphere[19–21] and the evolution of plants and animals[22,23]. Interestingly, since plants and animals colonised the land some 400 million years ago, oxygen has remained in what has become known as the 'fire window' between 16 and ~30% atmospheric oxygen, where simple calculations and models suggest that atmospheric oxygen levels at, or above 25% vol., may threaten the regeneration of drier forests following frequent fires, whilst 30% might be the upper limit for wet forests[7,24,25]. Above such high concentrations in the atmosphere the frequency and behaviour of fires would suppress vegetation so much that it would be impossible for large forest ecosystems to exist[25]. Regulation of atmospheric oxygen involves several sinks and sources that provide regulation against high and low levels of atmospheric oxygen.

The long-term source of oxygen to the atmosphere is the burial of organic carbon in sediments[24,26] (Fig. 1), because the burial of organic carbon, represents oxygen released by photosynthesis that has not been re-consumed by heterotrophs. At present, roughly half of the Earth's organic carbon for carbon burial comes from land and the other half from the ocean, but almost all is buried in ocean sediments[27]. In the ocean, phosphorus, not nitrogen is the limiting nutrient. Therefore, the abundance of phosphorus, and its link to productivity, determine how much carbon is buried in ocean sediments.

Under anoxic conditions in the ocean there is more available phosphorus as removal of phosphorus to iron oxyhydroxides ceases and phosphorus is preferentially recycled in low-oxygen conditions[24,28]. Increases in phosphorus result in enhanced productivity and increased organic carbon burial which prevents atmospheric oxygen from falling too low and increases the abundance of atmospheric oxygen, creating a negative feedback loop, which counteracts against low oxygen[28–31] (Fig. 1). Although this mechanism counteracts ocean anoxia, if oxygen concentration rises so that anoxic waters are removed from the ocean, this feedback switches off. Hence this mechanism cannot explain the regulation of the upper limit of oxygen concentration in the atmosphere[7,24].

One of the major sinks of atmospheric oxygen is through oxidative weathering of primarily old organic carbon, and iron pyrite[7] (i.e. rusting) however, based on its geological abundance in sediments, would be insufficient to be able to counteract extreme rises in atmospheric oxygen[7,32]. Oxidative weathering is believed to go almost to completion at present day atmospheric oxygen levels[7,33,34] and so has diminishing power as oxygen concentrations increase and cannot prevent oxygen rising[7,32] above that critical for the existence of large land vegetation. Therefore, processes that act on the land that connect to the rate of carbon burial in both terrestrial and ocean sediments to variations in atmospheric oxygen are required. Because of the co-dependence of fire and forest ecosystems with the abundance of oxygen in the atmosphere, fire potentially provides a highly sensitive link between oxygen concentration and the long-term atmospheric oxygen source from organic carbon burial.

It is suggested that high fire frequency and intensity regulate atmospheric oxygen by suppressing plant growth and biomass production on land, meaning that less organic material is available for carbon burial both on the land and via transport to the ocean[30,35]. One suggestion is that this is regulated by a negative feedback linked to phosphorus redistribution (Fig. 1a). Here in the Kump model[26] as atmospheric oxygen rises and fire becomes more frequent, phosphorus is redistributed from the land to the ocean, reducing the overall carbon-phosphorus burial ratio in the ocean (because marine organisms tend to have lower C:P ratios), thus reducing carbon burial and hence the oxygen source[7,26]. This negative feedback via phosphorus distribution is shown in Fig. 1a and measured data agree that fire shifts phosphorus to the ocean[36]. However, questions have been raised as to whether a decline in phosphorus on land reduces land organic carbon burial[7,37]. Another regulating mechanism via phosphorus weathering (Fig. 1b)[24] is suggested in Lenton-Watson[24] and the COPSE model[37]. Here increased fire frequency will tend to shift ecosystems from forests to faster-regenerating vegetation with lower biomass[7,11,38,39]. Thus, increases in atmospheric oxygen increase fire-frequency, lowering the overall total biomass of the planet. Indeed, it has long been suggested that very high levels of atmospheric oxygen >35% would mean that fires would return so frequently to landscapes that full regrowth of forest would be impossible[24]. Therefore, this decline in biomass in turn suppresses weathering of the limiting nutrient phosphorus by land plants (see Fig. 1b) from silicate rocks, which at present amplify weathering of rocks by up to an order of a magnitude[7,24]. The assumption in the COPSE model[24,37] is that the terrestrial biosphere never becomes P-limited and always takes up sufficient weathered P to meet its growth requirements. The remainder is transported to the ocean and hence therefore continues to act to regulate carbon burial on both the land and the ocean. In this case, as fire increases and land biomass decreases, P-weathering by the root action of plants and their associated mycorrhizal fungi is diminished, lowering the P source to the ocean. Hence both ocean productivity slows due to the lower delivery of P, suppressing C-burial in ocean sediments and fire's suppression of large land biomass lowering overall C-burial on the land.

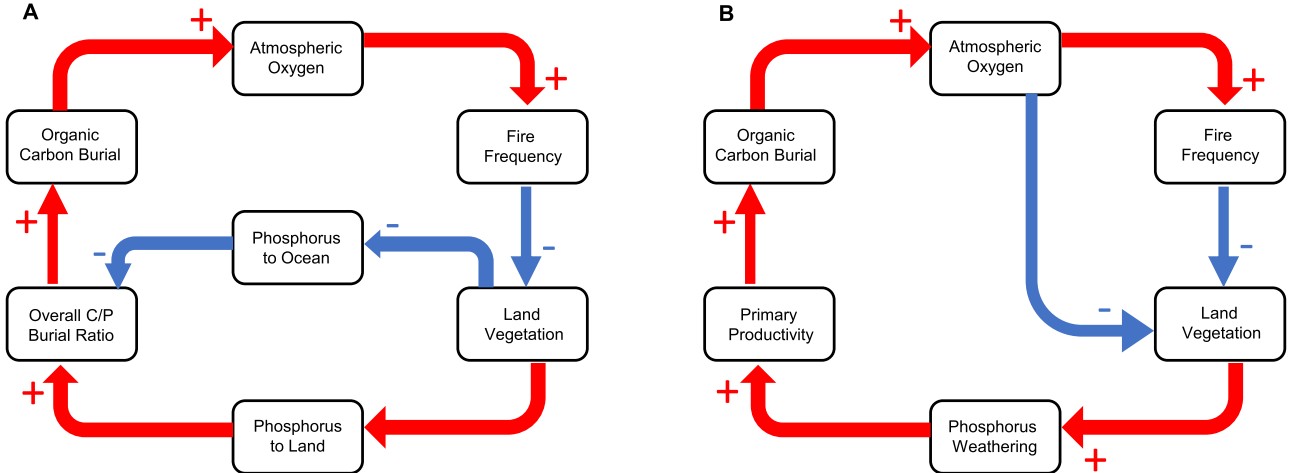

**Fig. 1 Negative fire feedbacks on atmospheric oxygen.** Feedback possibilities are via: **A** Phosphorus redistribution and **B** phosphorus weathering with the inclusion of the direct feedback of oxygen on plant productivity. The sign of each arrow represents the relationship between the two connecting boxes, with a (red) plus sign indicating a direct relationship (e.g. increasing oxygen increases fire frequency whilst decreasing oxygen decreases fire frequency) and a (blue) negative sign indicates an inverse relationship (e.g. increasing fire frequency decreases land vegetation whilst decreasing fire frequency increases land vegetation).

Biogeochemical models that run over Phanerozoic timescales (550 million years) are used to predict the history and controls on the abundance of oxygen in the atmosphere. All of these models have a representation of sedimentary reservoirs such as carbon, oxygen and sulfur, which are coupled to smaller, ocean-atmosphere reservoirs[40] and some like the Carbon, Oxygen, Phosphorus, Sulfur and Evolution (COPSE) model[37,40,41] include terrestrial primary productivity and basic fire feedbacks. Without fire feedbacks, models have predicted up to 35% vol. $O_2$ in the atmosphere[42,43]. However, by including negative feedbacks involving fires and phosphorus weathering and redistribution, this upper-limit in the model can be brought down to around 25% vol.[7,44]. For the COPSE model, negative feedbacks involving vegetation are needed for atmospheric oxygen to be stable at all. Where inclusion of a negative fire feedback not only suppresses extreme rising of atmospheric oxygen, but the upper-limit of oxygen depends on how strong the fire feedback is set to be.

Most biogeochemical modelling and proxy-based studies suggest that atmospheric oxygen ($pO_2$) rose to levels >24% vol. ~140 Ma during the Cretaceous[45–48], and fire-based proxy records imply a contemporaneous rise in fire-activity[17,18,48]. Shortly after $pO_2$ rose to super ambient levels, the first flowering plants (angiosperms) appeared on the planet (~135 Ma) and rapidly diversified by ~100 Ma[49–51]. Angiosperms appear to have acted as agents for expansive ecosystem change during the Cretaceous[52–54] owing to some key functional traits. It has been hypothesised that the estimated low leaf mass per area of the earliest angiosperms (135 Ma) added an easily dryable and ignitable fuel load that enhanced Cretaceous fire-frequency[15]. From around 100 Ma angiosperms are predicted to have evolved unrivaled $CO_2$ uptake and transpiration capabilities as evidenced by estimated enhanced leaf veination when compared to existing plant groups[54]; where it has been suggested that this gave angiosperms a considerable competitive advantage over slower-growing pteridophytes and gymnosperms, especially in a dark and disturbed environment[55]. This diversification of angiosperms has been suggested to have given rise to novel fire regimes that stimulated a positive feedback on their invasive capabilities in recently burned areas[14,15]. Flammability experiments combined with numerical models of fire behaviour suggest that the evolution of angiosperm shrubs and small trees (~100 Ma) enhanced the occurrence of rapidly spreading fires that burned with high intensity[14]. Therefore,

wildfires are expected to have been more widespread in the Cretaceous, both due to evolutionary driven changes in fuel and due to increases in atmospheric oxygen concentration.

In this paper we expand the COPSE (Carbon-Oxygen-Phosphorus-Sulphur-Evolution) model[37,56] to consider whether fuel-driven changes to fire activity during the Cretaceous had the ability counteract rising atmospheric oxygen at this time, thereby testing the ability of major events in the history of land plants to have played a strong role in regulating the abundance of oxygen in the atmosphere via their interaction with fire. COPSE includes the effects of fire-feedbacks on regulating atmospheric $pO_2$, but this is limited to imposing a chosen fire-feedback strength across the entirety of the Phanerozoic, producing a suite of $pO_2$ predictions according to either no fire-feedback, weak fire-feedbacks or strong fire-feedbacks being present[7]. Here we update the COPSE model to qualitatively capture the effect of changes to Cretaceous fire regimes based on evolutionary innovations in plants, fossil evidence for increased fire activity and high oxygen enhancement of fire. We reveal that a shift to a more flammable biosphere at this time appears to have resulted in a lowering of atmospheric oxygen and a strengthened negative feedback loop that has since prevented large swings in the abundance of oxygen in Earth's atmosphere. Hence, wildfire plays an essential role in maintaining our atmospheric composition.

## Results and discussion

**Diminishing fire suppression throughout the Cretaceous.** In order to examine the changes in biogeochemical cycling and atmospheric $O_2$ regulation under an evolving fuel scenario, we alter the fire suppression ratio ($f_{suppression}$) in the COPSE model from 135 Ma onwards according to major phases of innovation in angiosperms and their influence on fire regime. The $f_{suppression}$ parameter is a measure of the potential increase in terrestrial primary productivity under zero fire activity, i.e. setting $f_{suppression} = 10\%$ would be to assume that the terrestrial biosphere would be 10% more productive were there no fires. A larger value of $f_{suppression}$ represents a biosphere that is more vulnerable to fire. For the present day vegetation, a ~50% increase in productivity may be realised without wildfire[7]. We capture changes in vegetation during the Cretaceous and assess their potential fire feedbacks in the model according to the predicted angiosperm-driven fire shifts described in the following and which are summarised in

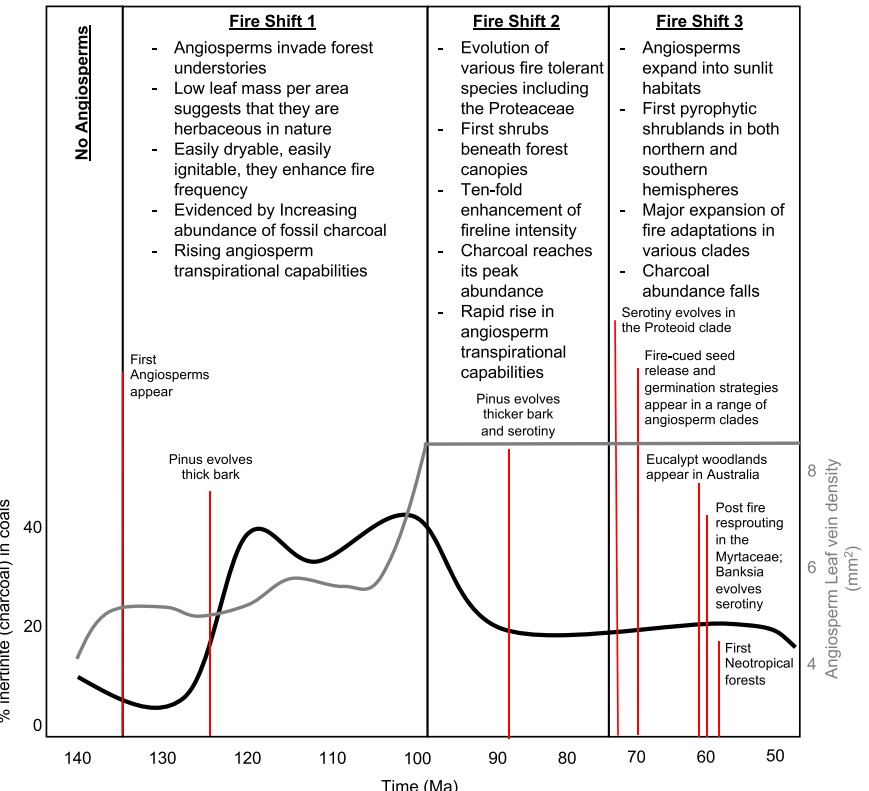

**Fig. 2 Cretaceous plant innovations and their linkage to fire regimes.** We describe three fire shifts-that we propose drive changes in fire suppression of biomass between 140 and 50 Ma. These shifts are informed by the key evolutionary changes and innovations listed in each box that are used to determine the fire suppression ratio in the COPSE model (see Fig. 3a). Specific events are marked by labelled red lines. This information is set against the abundance of inertinite (fossil charcoal) found in coals (black line) (data from ref. [47]) and changes in angiosperm leaf vein density (grey line) (data from ref. [54]) is shown highlighting the transition to plants with high transpiration capabilities.

Fig. 2. In these additions, we do not assume any changes in the modes of transport, or the bioavailability of weathered P. We assume that biotic weathering of silicates is present in all terrestrial systems both via root action and mediated by subsurface mycorrhizal fungi neither of which will be removed completely by fire, just suppressed, by the changes enacted in the model.

**Fire Shift 1—135 to 100 million years ago.** We assume that the earliest angiosperms appear ~135 Ma[51] and that they are of small stature and likely herbaceous in nature[55]. At this time fossil angiosperm leaves from Brazil, China, Europe, North and South America appear to have vein densities similar to Early Cretaceous ferns and gymnosperms and therefore likely had equal productivity potential[57,58]. However, estimates of leaf mass per area (LMA) from several North American fossil angiosperm floras reveal that the early angiosperms had low LMA, half that of gymnosperms, growing at the same time[59]. Based on the extant relationship between LMA and leaf economic traits this suggests that early angiosperms were likely fast-growing with short-lived leaves[59]. This would tend to make their leaves easily curable during the dry season making them highly flammable, where it has been suggested that an early angiosperm fire cycle was created similar to that of the modern grass-fire cycle of today[15].

These leaf traits would also tend to enhance the ease of surface fire spread in ecosystems where surface fuels were previously dominated by conifer litter[14]. Fern understories remained abundant during this period and have been shown to be capable of burning intensely and carrying fires with rapid spread[14]. Because $p$O$_2$ was tending to rise at this time[7] this could have had doubled fireline-intensity in both herbaceous and fern dominated

understories[14] leading to an increase in crown fire frequency[14] enhancing mortality in large land plant biomass. This shift in the fire appears to be supported by a rise in the abundance of fossil charcoals found in coals and sediments at this time[18,47] and links to the appearance of the first likely fire-adaptations in the Pinaceae family[13]. This family appeared some 240 million years ago but did not appear to develop their first fire adaptive traits until 126 Ma[13] where they evolved fire protective thick bark[13]. This further supports the idea that there was a change in the fire regime during this period and has also been suggested to be linked to the appearance of the angiosperms and rising atmospheric oxygen at this time.

**Fire Shift 2—100–75 million years ago.** From 100 Ma angiosperms were present as small trees and shrubs[54] and ferns began to diversify[57]. Monocots and early eudicots appeared between 100 and 90 Ma and were present in conifer forests of the Northern Hemisphere[60]. These included fine-leaved rhizomatous and cormaceoues families such as the fire-tolerant Iridaceae (103 Ma), Haemodoraceae (98 Ma) Asparagaceae (91 Ma)[61] and the Cyperaceae (88 Ma) and Poacea (89 Ma)[62,63]. Along with Ranunculaceae (~100 Ma)[64] and the Berberidaceae[13]. In addition, there is fossil evidence for open, Proteaceae dominated 'heathlands' in the Late Cretaceous of Australia and the emergence of sub family Proteoideae in Gondwana from 88 Ma[65]. The Proteaceae are a clade of typically schlerophylous shrubs that are well known for their fire-tolerant and fire adaptations. Fire behaviour models of this invasion of shrubs and fine-leaved plant morphotypes into forest understories has suggested that these new fuels were capable of carrying rapidly spreading and intense fires

with fireline-intensities at least 10 times greater than in forest understories previously dominated by conifer litter or herbaceous angiosperms[14]. These new fuels were capable of maintaining high-intensity fires even at relatively high fuel moisture contents leading to more destructive fires, owing to their ability to cause significant canopy scorch and a high probability of transitioning to crown fires[14]. Belcher and Hudspith[14] suggest that this created a fire regime that would have increased conifer mortality allowing angiosperms to invade at the expense of conifer tree forests[14,15].

The abundance of charcoals in coals and sediments peaks at ~100 Ma supporting a second enhancement of fire at this time which is followed by further adaptations in the Pinaceae appear around 89 Ma. Here, the evolution of thicker bark and fire cued seed release from serotinous cones evolves and has been suggested to be the result of the appearance of intense crown fires[13]. Such serotinuous traits occur in conifer trees that grow in regions dominated by crown fire regimes today[13,61]. As such between 100 and 75 Ma there is good reason to assume that there ought to have been less large land plant biomass on account of this shift in fire regime. The concomitant observable rapid increase in the vein density of angiosperm leaves from 100 Ma (Fig. 2), that it eclipses living and fossil non-angiosperm vascular plants during this time[54], suggests that Late Cretaceous angiosperms had leaves capable of very high gas exchange capacities[53,54] that ought to have assisted their ability to recolonize sites disturbed by fires more rapidly than conifers[14]. This highlights that the start of the Cenomanian (~100 Ma) likely saw a significant shift in fire regimes and was a major turning point in transpirational, productivity and atmospheric evolution.

**Fire Shift 3—75 to 50 million years ago**. Several clades of angiosperm that occur in modern fire-prone settings appear in the Late Cretaceous. Throughout this time period angiosperms expand into sunlit open habitats and therefore easily dryable habitats, forming pyrophytic shrublands. For example, Gondwanan Proteaceae moved out into open habitats around 88 Ma. The evolution of serotiny in the Proteoid clade of Proteaceae occurs not long after ~74 Ma shortly followed by the evolution of fire-cued seed release or fire germination strategies around 71 Ma in a range of angiosperms clades[65]. The evolution of fire adaptations, emerging in both the northern and southern hemispheres, throughout the Late Cretaceous is noted as being quite remarkable[65]. Eucalypt woodlands, well known for the flammability enhancing traits and their propensity to self-immolate[66] seem to have appeared in Australia around 62 Ma[67] and post-fire epicormic resprouting in Australian Myrtaceae (including Eucalypts) ~60 Ma[67]. At a similar time, the genus *Banksia* has been found to have evolved serotiny ~60.8 Ma[68]. As such this spread of pyrophytic ecosystems and fire-adaptations developing both north and south during this period suggests that the influence of fire on the Earth system must have been strong during the closing phases of the Cretaceous and into the Early Palaeogene.

**Impact of decreasing fire suppression on oxygen regulation**. Figure 3 details (a) the assumed suppression ratio of vegetation over time ($f_{suppression}$) based on the three phases of innovations in angiosperms and their associated shifts in fire, outlined above and indicated in Fig. 2 (b) the influence of this on land-derived organic carbon burial (one of the fire-based limitations on $O_2$ production), and (c) its influence on the predicted atmospheric $O_2$ mixing ratio between 150 and 50 Ma. The coloured area shows a sensitivity analysis to varying the suppression ratio ($f_{suppression}$), with all scenarios showing an increased susceptibility of the terrestrial biosphere to fires over the Cretaceous in line with the evolutionary changes detailed above. The dotted line shows the

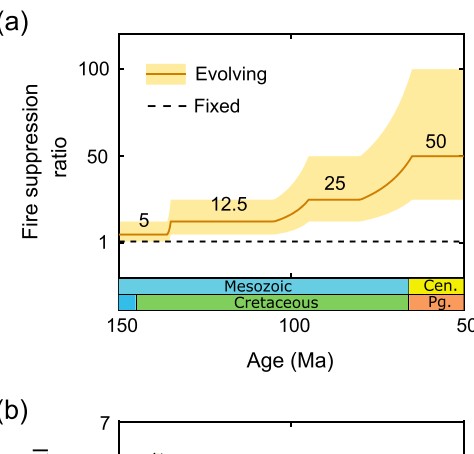

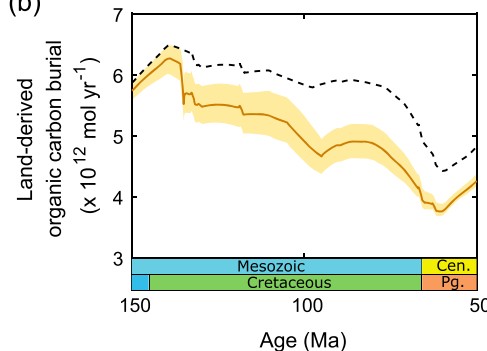

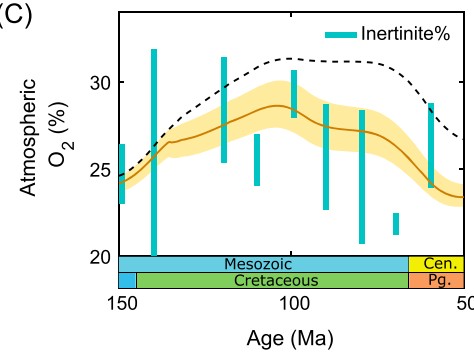

**Fig. 3 Model run from 150 to 50 Ma with changing fire feedbacks. a** Dark orange line with yellow shaded area shows an evolving 'fire suppression ratio', representative of the changing fuel source due to angiosperm evolution. The yellow shaded area indicates how uncertain the evolutionary changes in the fire regime are and how this is propagated to the model results. The size of the uncertainty window represents the suppression ratio of the previous and next evolutionary stages. Black dashed line shows fixed weak fire feedback in the baseline COPSE model. **b** Modelled land-derived organic carbon burial. **C** Modelled atmospheric oxygen mixing ratio compared to inertinite-derived estimates from ref. [47] as blue bars.

original COPSE model outputs in which fire suppression is assumed to be weak throughout.

The phased introduction of angiosperms and their influence on fire regimes throughout the Cretaceous acts to suppress global phosphorus weathering because the fires suppress long-lived large land biomass and C-burial on land. This decline in silicate weathering by plants diminishes the amount of phosphorus transferred to the ocean (where less carbon is buried per unit phosphorus), lowering the ocean based and overall organic carbon burial flux of the planet (Fig. 3b), which is the source of atmospheric oxygen (Fig. 1). This significantly lowers atmospheric oxygen levels from the mid to late Cretaceous (Fig. 3c), and compared to the model run with no evolving fire regimes we

can estimate that changes to biosphere flammability are responsible for around half of the $O_2$ reduction between the mid Cretaceous and Paleogene, indicating that evolutionary innovations in plants, coupled to fire, may have had the potential to both support the spread of angiosperms[14,15] and strengthen the regulation of the abundance of oxygen in Earth's atmosphere from the mid Cretaceous onwards. The oxygen rise over 150–110 Ma in both model simulations is driven by tectonic factors over this time where increased rates of erosion and subduction-degassing of carbonates promoted an increased input of phosphate from weathering[18] which influences carbon burial in the ocean (but not on the land).

**Potential effects of fire–oxygen feedbacks on the biosphere**. The predicted ~4% vol. lowering of $pO_2$ around the end of the Cretaceous from levels as high as ~30% (weak feedback) down to ~25% (evolving feedback) is significant because above 23% vol. $pO_2$ moist fuels are increasingly able to carry fire. For example, at 30% vol. $pO_2$ fires ought to be able to spread in fuels of moisture content ~112% (dry weight), according to the relationship $M_{ex} = 8O_x$-128 (where $M_{ex}$ is the moisture of extinction % dry weight and $Ox$ the % $pO_2$)[69], which contrasts sharply to modern day moisture of extinctions that are between 12–40% (20.9%) and 64% at 24% vol. $pO_2$. During the Cretaceous, $pO_2$ would likely have been too high to be favourable for the expansion of major closed canopy angiosperm tropical rainforests. This is intriguing because the majority of studies of leaf macrofossil assemblages suggest that closed canopy angiosperm tropical rainforests were not established until the Palaeocene[70–73], despite palaeoenvironmental observations and model estimates indicating locally persistent climatic conditions ought to have been able to support angiosperm tropical forest during the Late Cretaceous[74,75]. Whilst, angiosperm trees were present in the Late Cretaceous, these trees appear to have had small leaves, and are considered to have the typical morphology of woodland leaves, rather than that of closed tropical rainforest[76].

The earliest evidence of neotropical rainforest appears to be 58Ma[77]. By the end of the Cretaceous very high vein densities are observed in angiosperm fossil leaves[54] and are considered to have had high gas exchange capacities[78] (Fig. 3). These observations are significant because photosynthesis and transpiration by leaves are a primary influence on the cycling of water on land. Therefore evolutionary changes in the rate of gas exchange in leaves and variations in the rate of water vapour transfer between plants and the atmosphere should also have fed back to ecosystem flammability by influencing fuel moisture and would interact with the relationship between $pO_2$, the moisture of extinction and the spread of fire. This enhancement of transpiration rate[54] towards the end of the Cretaceous and into the Palaeocene should have enhanced fuel moisture in some plant communities (such as wetland or swampy settings, that were somewhat fire resistant) counteracting the $pO_2$ relationship to the moisture of extinction that would be requiring fuel to be drier for ignition, such that new transpirational capabilities might begin to reduce the risk of fire spread.

We suggest therefore that closed-canopy angiosperm tropical rainforests were not possible until fire feedbacks brought down $pO_2$ to 25% vol. and the evolution of very high leaf vein density had occurred in angiosperms (Fig. 3). Their ability to enhance rainfall and trap moisture beneath the dense forest canopy would increase fuel moisture, lowering flammability and increasing the ecosystem's resistance to fire even under the superambient $pO_2$ concentrations estimated for the time. We propose that the assembly of the closed canopy angiosperm tropical biome may therefore have been in part reliant on earlier angiosperm fire-

feedbacks that drove down $pO_2$ to concentrations more conducive to allow the establishment of pockets of tropical angiosperm forest. Once locally established these could then raise fuel moisture above the moisture of extinction required for fire to spread, thus creating a feedback that in turn allowed their expansion.

As well as lowering the stable $O_2$ level, increased terrestrial flammability throughout the Cretaceous should alter the strength of the feedback regime between plants and oxygen (Fig. 1). A perturbation to the Earth system, such as a tectonically driven change in the weathering-release of phosphorus from rocks, might be expected to alter productivity and cause a rise or fall in $O_2$. Indeed, such a perturbation, if large enough and not compensated for by changes to fossil carbon weathering, could stifle animal evolution by depleting atmospheric oxygen. A more flammable biosphere would be expected to increase the strength (or 'gain') of the negative feedback loop encompassing fire and plant productivity and would therefore damp any external perturbation (Fig. 1).

In order to test the importance of terrestrial flammability for the stability of $pO_2$, we ran perturbations of the COPSE model around the present-day steady state (Fig. 4 and Supplementary Information). In this analysis we increased the global delivery rate of the limiting nutrient phosphate by 50% for a 50 Myr period, and assessed the model response for a range of fire suppression ratios (1–50%) which represent biospheres that are either strongly (50%) or very weakly (1%) suppressed by fire activity over long timescales. This treatment is simplistic, as many global processes that increase phosphorus input would also alter the weathering rates of fossil organic carbon and other Earth system processes. Nevertheless, it is useful to demonstrate how the fire feedback strength alters the regulation of atmospheric $O_2$.

In the case where a wildfire has little effect on the terrestrial biosphere (dark red lines, Fig. 4), the increase in phosphate delivery results in a greater global rate of organic carbon burial and a rise in $pO_2$ from the present-day mixing ratio of 21% to around 32%. Although fire has minimal effects on the modelled vegetation, terrestrial NPP drops slightly due to the increased rates of photorespiration at high $pO_2$ assumed in the model, highlighting the increase in global carbon export and burial occurs in the marine realm. However, if it is assumed that wildfire has a significant suppressive effect on terrestrial vegetation (e.g. 50%, yellow line, Fig. 4), the model response is significantly damped. As $pO_2$ rises, fire severely limits terrestrial productivity, hence limiting the $pO_2$ rise by roughly a factor 2. Therefore, we conclude that the evolution towards a more highly flammable terrestrial biosphere has not only resulted in lower $pO_2$ (Fig. 3) but has strengthened a planetary negative feedback loop that prevents large swings in $pO_2$ (Fig. 4).

Using a simple box model, we are able to suggest that evolutionary innovations in plants over multimillion-year timescales likely have the capability to influence fire regimes and their feedback to the regulation of atmospheric oxygen. This simple approach should be built upon in further research if we are to understand the importance of long-term coupled plant-based regulatory processes on our planet. Future work should seek to develop spatially explicit dynamic global vegetation models capable of operating over macro-evolutionary timescales to allow estimates of the distribution of ancient plant groups across Earth's palaeoclimatic zones, which is a limitation of our model. Such a model would need to be coupled to a spatially explicit fire behaviour model that would allow the fire feedback to total land plant biomass to estimate the impact on P-weathering and redistribution required to influence long-term carbon burial and oxygen regulation. This would allow us to understand the role that the different biomes across climatic zones have likely played

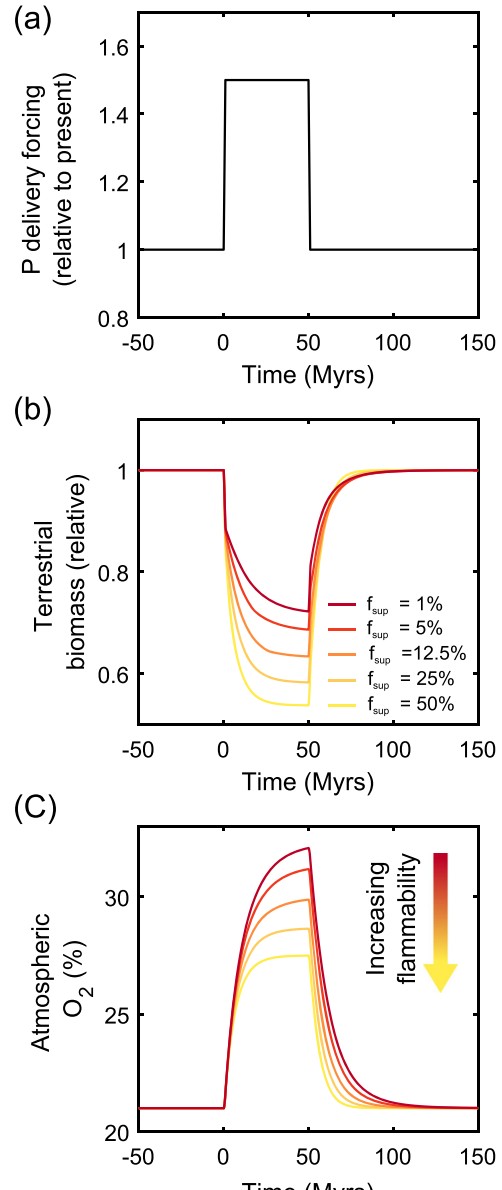

**Fig. 4 Model perturbation response for varied fire suppression ratios.** Suppression ratio is the assumed increase in global terrestrial NPP under the complete absence of fires, indicative of overall biosphere flammability. **a** Model is forced by increasing the delivery rate of nutrient phosphate by 50% for a 50 Myr period. **b** Relative terrestrial biomass. **c** Modelled atmospheric oxygen mixing ratio.

in the long-term regulation of Earth's atmospheric composition and potentially improve their management into the future.

In summary, it seems that angiosperms have driven several feedbacks throughout the Cretaceous that supported their success in becoming the most dominant plant group in the world. In doing so, the fire regimes that these evolutionary events created would have tended to suppress the overall vegetation biomass at the time. Over geologic timescales, we hypothesise that this reduced the organic carbon burial source of atmospheric oxygen, by suppressing plant-induced weathering of phosphorus from rocks and/or transferring phosphorus from the land to the ocean, where less carbon is buried per unit of phosphorus. This counteracted geological drivers of rising $pO_2$ such that it was in a steady decline throughout the Late Cretaceous and into the

Paleocene. Thus, the rise of angiosperms may have contributed to regulating the planet more tightly within habitable bounds.

This essential role of fire in regulating the abundance of oxygen in the atmosphere, should not be underestimated and draws attention to human mismanagement of fire over the past century. Modern humans have entirely altered ignition patterns, with some 84% of fires today being started by current anthropogenic activities[79], we have altered the connectivity of fuels in landscapes, species composition and fuel structure[80]. We have little understanding of to what extent we have disrupted critical fire feedbacks to the Earth system. As we move forward into what might be termed the Pyrocene, it is clear that our management of wildfires and ecosystems has to incorporate into its vision the fact that natural relationships between fire and ecosystems are themselves a resource that critically secure the long-term balance of the Earth system processes that maintain the air that we breathe.

## Methods
COPSE (Carbon, Oxygen, Phosphorus, Sulphur Evolution)[37] is a zero-dimensional long-term biogeochemical box model, which is based on the 'Redfield revisited'(Lenton and Watson 2000) and GEOCARB models[42,43]. We use the most recent iteration of the model, and full model equations are and the code and related files required to run the model are available in the Supplementary Information, Supplementary Data 1 and Source Data files. Fire feedbacks are represented by expressing the sensitivity of fire frequency to $pO_2$ and the impact that changes in fire activity have on vegetation biomass[24] and until now have not responded to fuel-driven changes in the fire. We update it here to allow for an evolving fuel scenario to capture additional fire forcings within the model. All model runs are for the full COPSE Phanerozoic run, however, we only alter fire feedbacks from 135 Ma onwards (see Supplementary Fig. 1). COPSE calculates relative terrestrial NPP by a linear combination of multiplicative factors, which describe the inhibition of primary productivity through photorespiration, growth rate dependence on $CO_2$ fertilization, and a parabolic relationship between temperature and productivity. To these is added a 'fire regulation' parameter, $f_{fire}$, dependent on the atmospheric $O_2$ mixing ratio as shown in Eqs. 1 and 2 below.

$$V_{npp} = f_{presp}(O_2) \times f_{fert}(CO_2) \times f_{temp}(T) \times f_{fire}(O_2) \quad (1)$$

where

$$f_{fire} = \frac{k_{fire}}{k_{fire} - 1 + ignit(O_2)} \quad (2)$$

Here ignit is a measure of fire probability ('ignition component') that increases linearly with $pO_2$ (from 1 at $pO_2 = 21\%$ atm to, e.g., 24.45 at $O_2 = 25\%$ atm). The parameter $k_{fire}$ controls how much fires suppress vegetation biomass in the model. This parameter is easiest to view in terms of the current fire suppression of the biosphere, in other words, how much biomass would be present in the biosphere if there were no fires. Setting ignit = 0 represents this case and we use this to define a new parameter for 'suppression ratio' based on the original model parameter $k_{fire}$ (Eq. 3):

$$f_{suppression} = \frac{k_{fire}}{k_{fire} - 1} \quad (3)$$

COPSE's 'weak fire-feedback' is defined by $k_{fire} = 100$, and translates into $f_{suppression} = {\sim}1.01$. Whereas the 'strong fire-feedback' ($k_{fire} = 20$) translates into $f_{suppression} = {\sim}1.05$. The model therefore assumes that the present biosphere would have 1% or 5% more biomass in the absence of fires. It is reasonably likely that this number may be much bigger for the present day, and has likely changed over time considerably. In this work, we test values for $1.01 < f_{suppression} < 2$, representing fire suppression factors of 1% (biosphere relatively unaffected by fires) to 50% (biosphere severely affected by fires). Following Lenton[7] we also amend the formulation for ignit, following a linear fit to the experimental curve for 10% moisture rather than 20%[17] (Eq. 4).

$$ignit = ki_1 \times O_2(\%) + ki_2 \quad (4)$$

where $ki_1 = 48$ and $ki_2 = -9.08$, and the function is bounded at the min and max of the experimental data.

The details of the evolutionary history of the fuels used to describe our changes in $f_{suppression}$ are in the main text. But we highlight here that COPSE is not a spatially explicit model hence no reference to plants biogeographic distribution is included. We use a range of fossil data sourced from across the globe to document key evolutionary innovations in plants throughout the Cretaceous that themselves provide a variety of forms of evidence for changes in fire regimes. These data, as noted in the main text, are drawn from observations of fossils and inferences from molecular phylogenetics and are not biased towards any one region or climatic zone. Our aim was to gather as much evolutionary information about the

appearance of novel fuel types that would alter fire regimes and/or the evolution of adaptive traits in plants that would imply a more flammable planet. These are summarised in Fig. 2.

Using these data, we detail a scenario where the value of $f_{suppression}$ changes based on the evolving biosphere (the addition of angiosperms to existing ecosystems) (see Fig. 3a): Between 150 and 140 Ma we assume that there were no angiosperms present and apply a biomass suppression ratio of 5% (this level is used pre-135 Ma as part of the full run back to 550 Ma, as we are not looking at vegetation innovations and fire prior to this period). We then take 135 Ma to be the earliest appearance of angiosperms on the planet, which we anticipate led to more frequent fires and more rapidly spreading fires (as outlined in Innovation phase 1) whilst the rising $pO_2$ at this time enhanced fire spread and intensity in new and existing fuel types[14]; as such between 135 and 100 Ma the model ramps up the suppression ratio to 12.5%. Between 100 and 75 Ma the suppression ratio is ramped up to 25% as angiosperms shrubs evolved that have been predicted to burn at high intensity and be capable of causing crown fires in conifer overstories[11] (as outlined in Innovation phase 2). The suppression ratio is further ramped up again to 50% by the end of the period (innovation phase 3)). In our sensitivity analysis we test suppression ratios up to 50% and down to 1%. (Fig. 4); in all cases, both the evolving scenarios and baseline scenarios return to 21% $O_2$ for the present day (Supplementary Fig. 1).

**Reporting summary**. Further information on research design is available in the Nature Research Reporting Summary linked to this article.

## Data availability

All data used in the research is either contained within the manuscript or has been made available in the supplementary information. This includes: The model background in Supplementary Notes 1 and 2. All model equations Supplementary Note 3. Model forcing factors and model species are provided in Supplementary Tables 1, 2 and 3. Source data are provided with this paper.

## Code availability

All model code as provided as MATLAB files in Supplementary Data 1 (which also includes a readme.txt file).

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

## Acknowledgements

This research was funded by a European Research Council Starter Grant (ERC-2013-StG-335891-ECOFLAM) awarded to C.M.B. B.J.W.M. acknowledges a University of Leeds Academic Fellowship and the UK Natural Environment Research Council (NE/S009663/1). A.J.W. acknowledges a Royal Society Research Professorship. R.V. acknowledges support from the University of Exeter and the Royal Society. T.M.L. acknowledges support from NERC (NE/N018508/1) and the Leverhulme Trust (RPG-2018-046).

## Author contributions

C.M.B. conceived the study. B.J.W.M. edited and ran the model. C.M.B., A.J.W., R.V. and S.J.B., discussed and outlined regulatory mechanisms in regard to fire and oxygen. C.M.B., B.J.W.M. and T.M.L. designed the model scenarios. All authors contributed to the conception of the study and discussed the results. All authors contributing to the writing and editing the paper.

## Competing interests
The authors declare no competing interests.
