## [Peer Review File · Nature Communications]

REVIEWER COMMENTS

Reviewer #1 (Remarks to the Author):

Belcher et al. explore the possible effects the origin and evolution of angiosperms on fire regime and atmospheric oxygen levels from Cretaceous to recent. They present a very nice summary of the way in which this evolution unfolded, and then impose assumed impacts of enhanced angiosperm fire suppression on the COPSE model to quantitatively evaluate the effects. They find that the imposed stepwise increase in fire suppression reduces phosphate weathering and terrestrial organic carbon burial, thereby reducing Cretaceous increases in atmospheric oxygen levels that otherwise occur in the model, driven by presumed tectonic effects.

I like the paper, especially the way that the detailed angiosperm evolutionary history is presented and then incorporated into the model. And I strongly support the focus on fire feedbacks as the main regulatory mechanism for atmospheric O₂ in the high O₂, fire prone world of the later Paleozoic, Mesozoic and Cenozoic: regulation has to be where the effects are most severe (terrestrial ecosystems), otherwise these ecosystems would not have survived. I do have one significant concern, and some minor suggestions, but feel that the paper should be published in Nature Communications.

The main concern I have is with how the model runs presented in Fig. 3 were initiated. Are these segments of a longer Phanerozoic run, or initiated at 150 m.y.? I suspect the latter, since both the standard run and the evolving feedback runs begin at about 25% O₂ at 150 m.y. But do they both end up at 21% for 0 m.y.? Or were they both initiated at steady state? How these models are initiated does make a big difference in their subsequent long-term evolution (as the problems associated with the initial BLAG model clearly showed). The authors should be clear about this, and consider showing the full simulations in the SI.

Minor comments:

bottom of p. 4, not sure whether the "from" should be "at" because there's not an interval of time being referred to here.

top of p. 5 "Cretaceous" is superscripted.

middle of p. 9: Having a hard time reconciling the text (paragraph beginning with "Specifically. . ." with the Fig. 3c. Fig. 3c seems to show O₂ at 30% (weak feedback) and 26% (evolving feedback) (or higher for both) at the end of the Cretaceous, whereas the text says "as high as ~28% (weak feedback) down to >24% (evolving feedback). Maybe the figure (runs) was updated but the text not?

Next sentence, extra opening parentheses between "pO₂" and "fires"

Near bottom of p. 10, no hyphen for ...ly adverbs

Also here, of course tectonically driven changes in weathering also affect the sink for O₂ associated with fossil carbon weathering, in reality AND in the model. So this wouldn't necessarily change O₂ levels much (depends on the details of how P weathering and fossil C weathering respond, C/P of the buried carbon relative to the weathered rock, etc.

In this same vein, I wonder whether the authors increased the sink for O₂ associated with fossil C weathering in their perturbation experiment that led to Fig. 4. (p. 11).

p. 11: "damped" instead of "dampened"

p. 11-12: the sentence starting "Humans have" is run-on.

Lee Kump, Penn State

Reviewer #2 (Remarks to the Author):

This paper presents the hypothesis that the rise of angiosperms during the Cretaceous Period, supplementing or replacing gymnosperms, led to a strong fire feedback on atmospheric O₂ regulation. Angiosperms, for reasons this reviewer did not entirely understand, are more susceptible to forest fires. Fire intensity and frequency is also a strong function of atmospheric pO₂ (a non-controversial finding). Hence, according to the authors, the advent of angiosperms around 135 Ma resulted in better atmospheric O₂ regulation and, eventually, to a long-term decrease of atmospheric O₂ from ~28% by volume at 100 Ma to ~23% at 50 Ma. The O₂ regulation occurs because of phosphorus (P) feedbacks. Burning of land plants transfers P to the oceans, where the C:P ratio in organisms is lower, and so less carbon is buried and less O₂ is produced. Land plants may also accelerate P weathering, so burning them down reduces the total P supply.

The paper is well written and well referenced. The authors test their hypothesis using the COPSE model, which is well designed for such a study. That said, the arguments are all a bit 'soft', as it is difficult to quantify many of the physical/biological processes and feedback processes that are important to the argument. This limitation, unfortunately, goes with the territory. Atmospheric O₂ regulation by fires, or by anything else, is simply not that easy to quantify. It's a geobiological system, not a geophysical or geochemical one. This reviewer prefers systems that include better quantitative constraints. However, that does not mean that the present study is not useful. Indeed, I learned a lot from the discussion and now have a better understanding of the processes regulating atmospheric pO₂.

Revealing my distance from the subject matter, I have no detailed criticisms or comments. The paper seems like a useful addition to the literature on this topic. I see no reason why it should not be published essentially as is.

You may reveal my name to the authors.

Jim Kasting

Reviewer #3 (Remarks to the Author):

This is a comprehensive and rigorous review of the fire feedback coupled innovatively to novel discussions about ecological shifts—discussed in ways that perhaps only this group could. In many respects, it is a review paper. Lots of setup is provided before the specifics of the models are laid out. But that's fine. No, it's great. Many readers will appreciate the diverse layers of context and the clear explanations of complex relationships—made even more complex by evolving over time.

I was ready to rubber stamp my okay on this, but as I read the paper a couple of times I realized that I wasn't quite getting the link to phosphorus, which lies at the heart of the story. I was looking for consistency in arguments and struggling. As the authors note, "Over geologic timescales we hypothesize [a] reduced organic carbon burial source of atmospheric oxygen, by suppressing plant-induced weathering of phosphorus from rocks and/or transferring phosphorus from the land to the ocean, where less carbon is buried per unit of phosphorus." Also, "the phased introduction of angiosperms and their associated influence on fire regimes throughout the Cretaceous suppresses phosphorus weathering and transfers phosphorus to the ocean (where less carbon is buried per unit phosphorus), lowering the overall organic carbon burial flux, which is the source of atmospheric oxygen." (I will note that some of authors' sentences are too long and unnecessarily complicated... One sentence is eight lines long, for example.)

Importantly in terms of the P discussion, the authors also note, "questions have been raised as to whether a decline in phosphorus on land reduced land organic carbon burial." In other words, maybe P wasn't limiting in most of these systems, and the authors' point is moot. Today, for example, nitrogen is thought (at least by some/many) to limit in temperate regions, with P being more critical in tropical regions. So there will be important spatial/climatic differences in response/sensitivity. Like all papers, there are some central assumptions that underlie much/most of the paper. I'd like to see this one better deconstructed.

We are told that part of this story is the feedback relationship to weathering, whereby suppression of terrestrial ecosystems leads to dramatic reduction in weathering by land plants. Such plant-coupled weathering has an order-of-magnitude amplification effect today. The corollary, we are told, is a redistribution of P from the land to the oceans. A big part of this argument is that plant-catalyzed weathering facilitates P availability of land, which may or may not be essential to the ecosystem, and by doing so reduces P delivery to the oceans. This consequence could simply reflect more P being trapped on land as biomass, right? In addition, however, the authors draw a specific connection to weathering.

I might be missing something, but I imagine terrestrial weathering that favors P mobilization as being critical for delivery to the oceans as well. Under lower terrestrial weathering in the models discussed here, how is P principally transported to the oceans, and critically, is it in a readily bioavailable form? Again, I might be missing something simple, but these details could be the mechanistic umbrella that stands above the central conclusions of the paper. In other words, I would very much appreciate seeing more-specific attention paid to the critical P parts of the time-varying story—keeping in mind that the full network of key relationships and feedbacks will not be easy for nonspecialist readers.

Finally, tectonic controls are discussed briefly as related to erosion and weathering impacts on P cycling, but it is not clear whether these tectonic relationships would preferentially impact P availability of land or the oceans or affect both equally—and thus the consequences for the global mean C/P ratio of buried biomass that lies at the foundation of the authors' story are not apparent.

I am quite confident the authors will have a thoughtful response, and I am optimistic that a revised manuscript will be suitable for publication Nat. Comm. I look forward to that!

Reviewer #4 (Remarks to the Author):

I find this an interesting master narrative about the role of wildfire controlling planetary biogeochemical cycles. The scope is necessarily big picture and deep time. The argument has merit, but the devil is in the details and this is where the paper flounders. The biggest issue is the naive treatment of global vegetation patterns and dynamics. The authors are trying to convey an important idea about a major evolutionary innovation of the angiosperms and the nexus with fire and the spread of 'tropical rainforest'. But this is quite decoupled from reality. Tropical rainforests are by definition in the humid regions of the tropics. Today there are great expanses of savanna in the tropics in the seasonally dry tropics. In the early Tertiary the biogeographic zones are more complicated as the world is ice-free, palms were growing in the high latitudes for example. Where were the the tropics then? This issue is alluded to by the authors comment:

"angiosperm rainforests were not established until the Palaeocene, despite palaeoenvironmental observations and model estimates indicating locally persistent climatic conditions ought to have been able to support angiosperm tropical forest during the Late Cretaceous. Whilst, angiosperm trees were present in the Late Cretaceous, these trees appear to have had small leaves, and are considered to have the typical morphology of woodland leaves, rather than that of closed forest."

We need much more information up front about the authors understanding of the palaeo bioclimate zonation, and critically the available evidence and location for reconstructed vegetation. Could it be that the available fossil sites just sampled a seasonal climate?

The authors drawn parallels with extant vegetation such as:

"Additionally, there is fossil evidence for open, Proteaceae dominated 'heathlands' in the Late Cretaceous of Australia and the emergence of sub family Proteoideae in Gondwana from 88 Ma. The Proteaceae are a clade of typically sclerophyllous shrubs that are well known for their fire-tolerant and fire adaptations."

It is entirely possible to have combustible vegetation that rarely burnt and poorly fire adapted so the claim is not necessarily true.

Another example of a loose parallel is:

"Fire tends to shift ecosystems from forests to faster-regenerating vegetation with lower biomass"

This is not true. Some high biomass forests are highly fire adapted. The issue is the ignitions density and

the return times of fire. Very frequently burnt vegetation can be low biomass but trees can still persist as is the case for savannas. On that point the grass story is brushed over, including the old claim the radiation was linked to declining CO₂ concentrations.

The core question is fire frequency and fire severity which is very difficult to objectively determine from the deep time fossil record. The authors never grapple with fire ignitions, which until the evolution of hominins was from lightning, and again the areas likely to burn, and the frequency at which they burn require the association of lightning with dry seasons (noting this structure is different under higher oxygen).

The authors do not handle the anthropogenic ignition factor very well writing:

“Humans have entirely altered ignition patterns, with some 84% of fires being started by people, we have altered the connectivity of fuels in landscapes, species composition and fuel structure.”

Indigenous people have been using fire since the late Pleistocene and there is a case that their skilful usage has presented biodiversity and the disruption by European colonisation of traditional fire practices has caused problem, so the issue is not people but culture.

Overall, I think the piece is interesting as thought piece that is an avowed speculative comment supported by a model. The diagrams need to be much better incorporated into the text, especially Figure 1 which is gnomic, the reader needs to be led through these feedbacks to really emphasise the point both in the text and the caption. The bold assumptions about ignitions and palaeo-biogeography need to be spelt out and some global maps would help ground this piece which at times is a stream of ‘what if’ and ‘could be’ qualified as proposals, hypotheses, assumptions. There are a lot of these sort of stories out there, they are fun and thought provoking but difficult to prove them. I notice the idea that the huge fires thought to be associated with the asteroid impact 66 million years ago promoted by Vivi Vajda did not get a look in.

In summary, proving big ideas like that outlined in this manuscript require far more targeted research than fairly simple heuristic models based on lots of assumptions, the most basic being a palaeo-biosphere with limited bioclimatic zonation. So better acknowledgement of caveats and direction for further work would strengthen this piece.

Reply to Reviewers

Reviewer #1 (Remarks to the Author):

Belcher et al. explore the possible effects the origin and evolution of angiosperms on fire regime and atmospheric oxygen levels from Cretaceous to recent. They present a very nice summary of the way in which this evolution unfolded, and then impose assumed impacts of enhanced angiosperm fire suppression on the COPSE model to quantitatively evaluate the effects. They find that the imposed stepwise increase in fire suppression reduces phosphate weathering and terrestrial organic carbon burial, thereby reducing Cretaceous increases in atmospheric oxygen levels that otherwise occur in the model, driven by presumed tectonic effects.

I like the paper, especially the way that the detailed angiosperm evolutionary history is presented and then incorporated into the model. And I strongly support the focus on fire feedbacks as the main regulatory mechanism for atmospheric O₂ in the high O₂, fire prone world of the later Paleozoic, Mesozoic and Cenozoic: regulation has to be where the effects are most severe (terrestrial ecosystems), otherwise these ecosystems would not have survived. I do have one significant concern, and some minor suggestions, but feel that the paper should be published in Nature Communications.

The main concern I have is with how the model runs presented in Fig. 3 were initiated. Are these segments of a longer Phanerozoic run, or initiated at 150 m.y.? I suspect the latter, since both the standard run and the evolving feedback runs begin at about 25% O₂ at 150 m.y. But do they both end up at 21% for 0 m.y.? Or were they both initiated at steady state? How these models are initiated does make a big difference in their subsequent long-term evolution (as the problems associated with the initial BLAG model clearly showed). The authors should be clear about this, and consider showing the full simulations in the SI.

This is a useful point and one we now clarify in the revision in the methods section. We state in the methods that all model runs are of the full COPSE Phanerozoic run and do indeed return to ~21% O₂ at present. Additionally, we now show the full run in the SI.

Minor comments:

bottom of p. 4, not sure whether the "from" should be "at" because there's not an interval of time being referred to here. Removed

top of p. 5 "Cretaceous" is superscripted. Changed

middle of p. 9: Having a hard time reconciling the text (paragraph beginning with "Specifically. . ." with the Fig. 3c. Fig. 3c seems to show O₂ at 30% (weak feedback) and 26% (evolving feedback) (or higher for both) at the end of the Cretaceous, whereas the text says "as high as ~28% (weak feedback) down to >24% (evolving feedback). Maybe the figure (runs) was updated but the text not? Adjusted as noted

Next sentence, extra opening parentheses between "pO₂" and "fires" – removed extra parentheses

Near bottom of p. 10, no hyphen for ...ly adverbs removed the hyphen between tectonically

driven

Also here, of course tectonically driven changes in weathering also affect the sink for O₂ associated with fossil carbon weathering, in reality AND in the model. So this wouldn't necessarily change O₂ levels much (depends on the details of how P weathering and fossil C weathering respond, C/P of the buried carbon relative to the weathered rock, etc).

We have added “and not compensated for by changes to fossil carbon weathering” to address the reviewer’s point here.

In this same vein, I wonder whether the authors increased the sink for O₂ associated with fossil C weathering in their perturbation experiment that led to Fig. 4. (p. 11).

We did not, and we have now made this clear in the text, noting that although the scenario is simplistic, it is still a useful way to demonstrate how the fire feedback strength alters the regulation of atmospheric O₂.

p. 11: "damped" instead of "dampened" - changed

p. 11-12: the sentence starting "Humans have" is run-on. This is correct it is meant to be part of that paragraph

Lee Kump, Penn State

Reviewer #2 (Remarks to the Author):

This paper presents the hypothesis that the rise of angiosperms during the Cretaceous Period, supplementing or replacing gymnosperms, led to a strong fire feedback on atmospheric O₂ regulation. Angiosperms, for reasons this reviewer did not entirely understand, are more susceptible to forest fires. Fire intensity and frequency is also a strong function of atmospheric pO₂ (a non-controversial finding). Hence, according to the authors, the advent of angiosperms around 135 Ma resulted in better atmospheric O₂ regulation and, eventually, to a long-term decrease of atmospheric O₂ from ~28% by volume at 100 Ma to ~23% at 50 Ma. The O₂ regulation occurs because of phosphorus (P) feedbacks. Burning of land plants transfers P to the oceans, where the C:P ratio in organisms is lower, and so less carbon is buried and less O₂ is produced. Land plants may also accelerate P weathering, so burning them down reduces the total P supply.

The paper is well written and well referenced. The authors test their hypothesis using the COPSE model, which is well designed for such a study. That said, the arguments are all a bit ‘soft’, as it is difficult to quantify many of the physical/biological processes and feedback processes that are important to the argument. This limitation, unfortunately, goes with the territory. Atmospheric O₂ regulation by fires, or by anything else, is simply not that easy to quantify. It’s a geobiological system, not a geophysical or geochemical one. This reviewer prefers systems that include better quantitative constraints. However, that does not mean that the present study is not useful. Indeed, I learned a lot from the discussion and now have a better understanding of the processes regulating atmospheric pO₂.

Revealing my distance from the subject matter, I have no detailed criticisms or comments.

The paper seems like a useful addition to the literature on this topic. I see no reason why it should not be published essentially as is.
You may reveal my name to the authors.
Jim Kasting

Reviewer #3 (Remarks to the Author):

This is a comprehensive and rigorous review of the fire feedback coupled innovatively to novel discussions about ecological shifts—discussed in ways that perhaps only this group could. In many respects, it is a review paper. Lots of setup is provided before the specifics of the models are laid out. But that's fine. No, it's great. Many readers will appreciate the diverse layers of context and the clear explanations of complex relationships—made even more complex by evolving over time.

I was ready to rubber stamp my okay on this, but as I read the paper a couple of times I realized that I wasn't quite getting the link to phosphorus, which lies at the heart of the story. I was looking for consistency in arguments and struggling. As the authors note,

“Over geologic timescales we hypothesize [a] reduced organic carbon burial source of atmospheric oxygen, by suppressing plant-induced weathering of phosphorus from rocks and/or transferring phosphorus from the land to the ocean, where less carbon is buried per unit of phosphorus.”

We have left this sentence as it is as we believe that with all the former adjustments and additions (see answers to queries below) that this now provides a relevant summary that would make sense.

Also, “the phased introduction of angiosperms and their associated influence on fire regimes throughout the Cretaceous suppresses phosphorus weathering and transfers phosphorus to the ocean (where less carbon is buried per unit phosphorus), lowering the overall organic carbon burial flux, which is the source of atmospheric oxygen.”

We have revised this to provide more explanation as follows (and have also explained this more fully in the introduction as noted below).

“The phased introduction of angiosperms and their influence on fire regimes throughout the Cretaceous acts to suppress global phosphorus weathering because the fires suppress long-lived large land biomass and C-burial on land. This decline in silicate weathering by plants diminishes the amount of phosphorus transferred to the ocean (where less carbon is buried per unit phosphorus), lowering the ocean based and overall organic carbon burial flux of the planet (figure 3b); which is the source of atmospheric oxygen (figure 1).”

(I will note that some of authors' sentences are too long and unnecessarily complicated... One sentence is eight lines long, for example.)

We have aimed to improve sentences and better provide better explanations.

Importantly in terms of the P discussion, the authors also note, “questions have been raised as to whether a decline in phosphorus on land reduced land organic carbon burial.” In other words, maybe P wasn’t limiting in most of these systems, and the authors’ point is moot. Today, for example, nitrogen is thought (at least by some/many) to limit in temperate regions, with P being more critical in tropical regions. So there will be important spatial/climatic differences in response/sensitivity. Like all papers, there are some central assumptions that underlie much/most of the paper. I’d like to see this one better deconstructed.

The reviewer is correct that decreasing P availability on land may not reduce carbon burial in all ecosystems. That said, we continue to note below that sentence that even if carbon burial is not altered by changes to terrestrial P availability, it will be lowered by the direct action of fires, which are shown to reduce overall biomass. We have made this clearer and have now edited the sections in the introduction to explain these regulatory mechanisms in more detail.

We are told that part of this story is the feedback relationship to weathering, whereby suppression of terrestrial ecosystems leads to dramatic reduction in weathering by land plants. Such plant-coupled weathering has an order-of-magnitude amplification effect today. The corollary, we are told, is a redistribution of P from the land to the oceans. A big part of this argument is that plant-catalyzed weathering facilitates P availability of land, which may or may not be essential to the ecosystem, and by doing so reduces P delivery to the oceans. This consequence could simply reflect more P being trapped on land as biomass, right? In addition, however, the authors draw a specific connection to weathering.

We regret that this part of the paper was not clearly worded. The redistribution of P to the ocean is assumed to be due to burial of less organic carbon (and therefore organic P) on land. It is not a corollary of the change in biotic weathering. We have added the following in:

“The assumption in the COPSE model^{5,6} is that the terrestrial biosphere never becomes P-limited and always takes up sufficient weathered P to meet its growth requirements. The remainder is transported to the ocean and hence therefore continues to act to regulate carbon burial on both the land and the ocean. In this case as fire increases and land biomass decreases P-weathering by the root action of plants and their associated mycorrhizal fungi is diminished, lowering the P source to the ocean. Hence both ocean productivity slows due to the lower delivery of P suppressing C-burial in ocean sediments and fire’s suppression of large land biomass lowering overall C-burial on the land.”

I might be missing something, but I imagine terrestrial weathering that favors P mobilization as being critical for delivery to the oceans as well. Under lower terrestrial weathering in the models discussed here, how is P principally transported to the oceans, and critically, is it in a readily bioavailable form?

We do not assume any changes in the modes of transport, or the bioavailability of weathered P. Biotic weathering is present in all terrestrial systems, is often mediated by subsurface mycorrhizal fungi rather than plants themselves, and will not be removed completely by fire, just suppressed. We have also made this clearer in the revision. We have added the following text: “In these additions we do not assume any changes in the modes of transport, or the bioavailability of weathered P. We assume that biotic weathering of silicates is present in all terrestrial systems both via root action and mediated by subsurface mycorrhizal fungi neither of which will not be removed completely by fire, just suppressed, by the changes enacted in the model.”

Again, I might be missing something simple, but these details could be the mechanistic umbrella that stands above the central conclusions of the paper. In other words, I would very much appreciate seeing more-specific attention paid to the critical P parts of the time-varying story—keeping in mind that the full network of key relationships and feedbacks will not be easy for nonspecialist readers.

We really appreciate these comments and regret that we did not explain the debate around P sources and effects towards a broader audience. We have revised our introduction to much better explain these ideas and to state them more carefully and believe the paper is strengthened as a result. This section is highlighted in the text so that revisions can be seen.

Finally, tectonic controls are discussed briefly as related to erosion and weathering impacts on P cycling, but it is not clear whether these tectonic relationships would preferentially impact P availability of land or the oceans or affect both equally—and thus the consequences for the global mean C/P ratio of buried biomass that lies at the foundation of the authors' story are not apparent.

To answer the reviewer's question, tectonically-driven increases (or decreases) in P weathering impact only the marine biosphere. This means that the C/P ratio of global biomass is indeed altered, but the overall P source and sink has also changed. The C/P ratio is only critical if there is a fixed amount of P to go around (e.g. constant weathering).

We have overall better explained the section describing the two different model approaches to regulating high atmospheric oxygen for P- redistribution or P-weather but in respect to this comment as noted above we have added the following:

“The assumption in the COPSE model^{5,6} is that the terrestrial biosphere never becomes P-limited and always takes up sufficient weathered P to meet its growth requirements. The remainder is transported to the ocean and hence therefore continues to act to regulate carbon burial on both the land and the ocean. In this case as fire increases and land biomass decreases P-weathering by the root action of plants and their associated mycorrhizal fungi is diminished, lowering the P source to the ocean. Hence both ocean productivity slows due to the lower delivery of P suppressing C-burial in ocean sediments and fire's suppression of large land biomass suppresses overall C-burial on the land.”

And also added the following in the discussion to clarify this:

“The oxygen rise over 150-110 Ma in both model simulations is driven by tectonic factors over this time where increased rates of erosion and subduction-degassing of carbonates promoted an increased input of phosphate from weathering¹⁸ which influences carbon burial in the ocean (but not on the land).”

I am quite confident the authors will have a thoughtful response, and I am optimistic that a revised manuscript will be suitable for publication Nat. Comm. I look forward to that!

Reviewer #4 (Remarks to the Author):

I find this an interesting master narrative about the role of wildfire controlling planetary

biogeochemical cycles. The scope is necessarily big picture and deep time. The argument has merit, but the devil is in the details and this is where the paper flounders. The biggest issue is the naïve treatment of global vegetation patterns and dynamics. The authors are trying to convey an important idea about a major evolutionary innovation of the angiosperms and the nexus with fire and the spread of ‘tropical rainforest’. But this is quite decoupled from reality.

Tropical rainforests are by definition in the humid regions of the tropics. Today there are great expanses of savanna in the tropics in the seasonally dry tropics. In the early Tertiary the biogeographic zones are more complicated as the world is ice-free, palms were growing in the high latitudes for example. Where were the the tropics then?

This issue is alluded to by the authors comment:

“angiosperm rainforests were not established until the Palaeocene, despite palaeoenvironmental observations and model estimates indicating locally persistent climatic conditions ought to have been able to support angiosperm tropical forest during the Late Cretaceous. Whilst, angiosperm trees were present in the Late Cretaceous, these trees appear to have had small leaves, and are considered to have the typical morphology of woodland leaves, rather than that of closed forest.”

We regret the misunderstanding of ‘tropical rainforest’ – we are not describing tropical floras in general, other plants (e.g. conifers) have dominated the tropics in the past (during the Jurassic). We are talking about the evolution of a specific biome; that of closed canopy angiosperm tropical rainforest that has specific plant traits that make it characteristic of the shift towards our modern neotropical forest realm. Hence our suggestions in regard to closed canopy angiosperm tropical rainforest has nothing to do with biogeographic and climatic zonation as such, but rather the coupling of environmental conditions with evolving plant traits that together are capable of building a new biome.

We have added clarification in the text by using the term ‘closed canopy angiosperm tropical rainforest’ or where relevant neotropical rainforest (which refers to closed canopy angiosperm rainforest in the neotropical realm, specifically), to avoid misunderstanding.

We need much more information up front about the authors understanding of the palaeo bioclimate zonation, and critically the available evidence and location for reconstructed vegetation. Could it be that the available fossil sites just sampled a seasonal climate?

We respectfully disagree with this point. The fossil data and references used to provide the basis for our evolving fuel scenario as described in Figure 2 and are fully described in pages 6, 7 and 8 include data from across the globe. Therefore, none of the literature review information that we have used to develop our evolving fuel scenario are taken from a single specific climate zone. All evidence is fully referenced. Additionally, the COPSE model is not spatially explicit, and therefore does not require a palaeoclimatic biozonation. Hence, there is no spatially explicit reconstructed vegetation. This was described in in the SI (section 2 under Model Structure)

To clarify both these points we have added the following paragraph to the methods section of the main manuscript and improved the figure caption for Figure 2 to make better links to the model.

‘The details of the evolutionary history of the fuels used to describe our changes in $f_{suppression}$ are in the main text. But we highlight here that COPSE is not a spatially explicit model hence no reference to plants biogeographic distribution is included. We use a range of fossil data sourced from across the globe to document key evolutionary innovations in plants throughout the Cretaceous that themselves provide a variety of forms of evidence for changes in fire regimes. These data, as noted in the main text, are drawn from observations of fossils and inferences from molecular phylogenetics and are not biased towards any one region or climatic zone. Our aim was to gather as much evolutionary information about the appearance of novel fuel types that would alter fire regimes and/or the evolution of adaptive traits in plants that would imply a more flammable planet. These data are summarised in figure 2.’

The authors drawn parallels with extant vegetation such as:

“Additionally, there is fossil evidence for open, Proteaceae dominated ‘heathlands’ in the Late Cretaceous of Australia and the emergence of sub family Proteoideae in Gondwana from 88 Ma. The Proteaceae are a clade of typically sclerophyllous shrubs that are well known for their fire-tolerant and fire adaptations.” It is entirely possible to have combustible vegetation that rarely burnt and poorly fire adapted so the claim is not necessarily true.

We regret any misunderstanding here – in this section of the manuscript we are describing the evidence for changes in plants that link to fire. i.e. this section describes the plant groups that evolved throughout the Cretaceous. We are not drawing parallels here to any modern flora, these are actual fossil data and molecular dates for the appearance of these groups in the past. We also note that it has been indicated that the Proteaceae evolved fire cued seed release by 74Ma (as noted in this paragraph) (see Lamont and He, 2012). We take this as evidence that this clade was necessarily fire prone during the Cretaceous, as it is today.

Another example of a loose parallel is:

“Fire tends to shift ecosystems from forests to faster-regenerating vegetation with lower biomass”

This is not true. Some high biomass forests are highly fire adapted. The issue is the ignitions density and the return times of fire. Very frequently burnt vegetation can be low biomass but trees can still persist as is the case for savannas.

We have made no assumptions in regard to fire adaptations in this sentence or section of the manuscript. So the comment in regard to “some high biomass forests are highly fire adapted’ does not make sense to us. However, yes we agree that long-term survival of forest ecosystems is based on fire frequency and fire return intervals, and this is what we mean by “Fire tends to shift ecosystems from forests to faster-regenerating vegetation with lower biomass”. We follow this sentence with “Thus, increases in atmospheric oxygen increases fire-frequency...”. This is the point we are trying to make that, increasing oxygen above a significant point (i.e. 35% pO₂ Watson et al., 2000) will prevent any ecosystem from returning to forest because fire will return to the same site too often to allow the full regrowth of forest. We have however, clarified this – by re-writing the statement to be more explicit as follows:

“Increased fire frequency will tend to shift ecosystems from forests to faster-regenerating vegetation with lower biomass^{7, 17, 43}. Thus, increases in atmospheric oxygen increases fire-frequency, subsequently shifting ecosystems to lower productivity and biomass. Indeed, it has

long been suggested that very high levels of atmospheric oxygen >35% would mean that fires would return so frequently to landscapes that full regrowth of forest would be impossible⁵. Therefore, this decline in biomass in turn suppresses weathering by land plants (see Figure 1), which at present amplify weathering of silicate rocks by up to an order of a magnitude^{5, 7}.

On the point that ‘the grass story is brushed over, including the old claim the radiation was linked to declining CO₂ concentrations’. We do not mention grass because we have not assessed oxygen levels or fire feedbacks for the time period in which we see the spread of C₃ grasses into open habitats (i.e. grasslands) or C₄ grasslands (to which the reviewer alludes to). Our analysis stops at 50Ma. Open habitat C₃ grassland emerge at the earliest 40 Ma (Stromberg, 2011), with the majority appearing ~ 20Ma (Stromberg, 2011). The evolution of the C₄ photosynthetic pathway dates from the Oligocene (~34Ma) but C₄ expansion is not until the Miocene (~8Ma) (Stromberg, 2011).

The core question is fire frequency and fire severity which is very difficult to objectively determine from the deep time fossil record. The authors never grapple with fire ignitions, which until the evolution of hominins was from lightning, and again the areas likely to burn, and the frequency at which they burn require the association of lightning with dry seasons (noting this stricture is different under higher oxygen).

The ‘ignition component’ in COPSE captures how easy or difficult it is to start a fire (with lightning). This ignition component equation is taken from the US National Fire Danger Rating System, which has been modified to include O₂ hence this is a well-used baseline for ignition. We note that the model is fully outlined in the SI. COPSE does not have a built in spatially explicit lightning algorithm. Rather it used a probability of ignition scaling (see SI) that describes variations in the probability of ignition based on O₂ which links to the equations for $f_{suppression}$. This is detailed in the methods and the SI and in the papers that describe the COPSE model in full that are fully referenced. In regard to fire severity – we qualitatively describe key variations in estimated changes in fire regime, both frequency and behaviour, in our rationale for varying the feedback in the model. These are described on pages 6, 7 and 8 and in Figure 2. We have improved the figure caption for figure 2 and added a short section in the methods (which is described below in answer to another query).

The authors do not handle the anthropogenic ignition factor very well writing:

“Humans have entirely altered ignition patterns, with some 84% of fires being started by people, we have altered the connectivity of fuels in landscapes, species composition and fuel structure.”

Indigenous people have been using fire since the late Pleistocene and there is a case that their skilful usage has presented biodiversity and the disruption by European colonisation of traditional fire practices has caused problem, so the issue is not people but culture.

In this part of the manuscript we simply make note that we have shown how important fire is for regulating O₂ and we are indicating here that humans may have disrupted this regulation. But to improve clarity we have edited the following sentence “Humans have entirely altered ignition patterns, with some 84% of fires today being started by people” to

‘Modern humans have entirely altered ignition patterns, with some 84% of fires today being started by anthropogenic activities.....’

Overall, I think the piece is interesting as thought piece that is an avowed speculative comment supported by a model.

The diagrams need to be much better incorporated into the text, especially Figure 1 which is gnomic, the reader need to be led through these feedbacks to really emphasis the point both in the text and the caption.

We have improved the caption and made it clearer as to how this figure links to the text and the model.

The bold assumptions about ignitions and palaeo-biogeography need to be spelt out and some global maps would help ground this piece which at times is a stream of ‘what if’ and ‘could be’ qualified as proposals, hypotheses, assumptions. There are a lot of these sort of stories out there, they are fun and thought provoking but difficult to prove them.

We have explained how the ignitions works in COPSE above and the equations required to run the model are fully detailed in the SI. The ignition equation is actually derived from the US National Fire Danger Rating System so it is incorrect that this be termed a ‘bold assumption’. As noted above, COPSE is not spatially explicit as is clear from the SI description of the model, hence palaeo-biogeographic assumptions are not relevant and we have already noted above that the fossil data used here does not only sample a single climatic or biogeographic zone. We have added the following section in the methods to highlight this – as follows:

“The details of the evolutionary history of the fuels used to describe our changes in $f_{suppression}$ are in the main text. But we highlight here that COPSE is not a spatially explicit model hence no reference to plants biogeographic distribution is included. We use a range of fossil data sourced from across the globe to document key evolutionary innovations in plants throughout the Cretaceous that themselves provide a variety of forms of evidence for changes in fire regimes. These data, as noted in the main text, are drawn from observations of fossils and inferences from molecular phylogenetics for example and are not biased towards any one region or climatic zone. Our aim was to gather as much evolutionary information about the appearance of novel fuel types that would alter fire regimes and/or the evolution of adaptive traits in plants that would imply a more flammable planet. These data are summarised in figure 2.”

Finally, we disagree that there are “lots of the sort of stories out there”. No-one has sought to address how evolutionary changes in plants, the fuel for fires might link to regulation of atmospheric oxygen. To say that ‘they are fun’ disrespects a large volume of Earth System thinking upon which over decades to come more complex models will be built.

I notice the idea that the huge fires thought to be associated with the asteroid impact 66 million years ago promoted by Vivi Vajda did not get a look in.

We are interested in long term shifts not single events. Additionally, it was Wendy Wolbach (1989) that suggested the wildfires and some of the authors have themselves work extensively on this topic. It is not included as this is a single event that is not long lived enough to impose the relevant shifts in atmospheric oxygen.

In summary, proving big ideas like that outlined in this manuscript require far more targeted research than fairly simple heuristic models based on lots of assumptions, the most basic being a palaeo-biosphere with limited bioclimatic zonation.

We respectfully disagree. Both kinds of research are required. Complex and precise techniques like the reviewer suggests are born from broader initial studies that look into feasibility and fluxes on a global scale.

REVIEWERS' COMMENTS

Reviewer #1 (Remarks to the Author):

I feel that the authors have nicely addressed my concerns, and to the best of my knowledge, those of the other reviewers.

Reviewer #3 (Remarks to the Author):

I have read through the revised paper and, in particular, the responses to my comments and those from the other reviewers. The revisions and responses are detailed and effective. I am convinced a better, potentially more impactful paper has resulted. I am ready to recommend publication without delay.

Tim Lyons

Reviewer #4 (Remarks to the Author):

The authors have improved the manuscript and the assumptions underlying their argument have become clearer. Nonetheless major two assumption need to be further highlighted in the manuscript as they are a pathway to test the hypothesis presented in this paper. The first is the assumption that single vegetation types dominated the Earth during the rise of the angiosperms, sequentially replacing each other between 140 and 50 million years ago. This is obviously a simplifying assumption as the Earth has always had climate variation and biogeographic patterns. The second assumption is that the early angiosperms were 'weedy' and highly flammable and that eventually fire resistant 'rainforests' developed. This assumption is debated amongst experts. Feild et al (2011) argue that the basal angiosperms (fossil and extant) have low vein density like understory species whereas weedy species have high vein density. More research can illuminate if these assumptions are reasonable or not, it is not possible to be certain with the available evidence in hand.

Feild, T.S., Upchurch Jr., G.R., Chatelet, D.S., Brodribb, T.J., Grubbs, K.C., Samain, M.-S., Wanke, S. (2011) Fossil evidence for low gas exchange capacities for Early Cretaceous angiosperm leaves. *Paleobiology*, 37 (2), pp. 195-213. DOI: 10.1666/10015.1

Some minor points

Line 22 the term 'tropical rainforest' here precedes the subsequent qualification of this term

Line 38 some wrong with this sentence

Line 94 'fire' is used loosely clarify if this frequency, severity or both

Line 95 'suppression plant growth' add hence biomass production

Line 136 is the claim that the angiosperms thrived in both dark and disturbed environments or dark, disturbed environments?

Line 152 stressed the assumption that flammability drove angiosperm evolution – this is not certain.

Line 182 clarify you mean fire frequency

Reply to REVIEWERS' COMMENTS

Reviewer #1 (Remarks to the Author):

I feel that the authors have nicely addressed my concerns, and to the best of my knowledge, those of the other reviewers.

Many thanks

Reviewer #3 (Remarks to the Author):

I have read through the revised paper and, in particular, the responses to my comments and those from the other reviewers. The revisions and responses are detailed and effective. I am convinced a better, potentially more impactful paper has resulted. I am ready to recommend publication without delay.

Tim Lyons

Many thanks

Reviewer #4 (Remarks to the Author):

The authors have improved the manuscript and the assumptions underlying their argument have become clearer. Nonetheless major two assumption need to be further highlighted in the manuscript as they are a pathway to test the hypothesis presented in this paper. The first is the assumption that single vegetation types dominated the Earth during the rise of the angiosperms, sequentially replacing each other between 140 and 50 million years ago. This is obviously a simplifying assumption as the Earth has always had climate variation and biogeographic patterns.

We disagree that we have suggested that there are single vegetation types dominating the Earth. This manuscript draws on evidence for changes in the evolution of major plant groups and how these likely influenced broad scale fire regimes. We also use evidence from fossil plants and phylogenetic studies to suggest evidence for shifting fire regimes; such as the evolution of fire adaptations in various groups. We use these information to provide a rationale for shifting the COPSE model's fire feedback at certain points during the Cretaceous. We note that the COPSE model, does include CO₂ and temperature (i.e. climate) as drivers of NPP and weathering and therefore whilst, this is a simplified biogeochemical box model it does account for climate effects on total biomass production and also fire feedbacks to this.

Therefore, the issue with us suggesting this as an assumption is because we have not assumed this. We do not at any point say or believe that single plant types dominated the Earth during the rise of the angiosperms nor do these sequentially replace one another. Angiosperms and Gymnosperms are not plant types, they are entire high order clades that contain many genera and species.

In our manuscript we draw on palaeontological evidence for changes in various different plant families and genera that sit within these two clades and that of the Pteridosperms (e.g. ferns). At no point do we say these sequentially replaced each other, we say that the addition of angiosperms to Earth's ecosystems altered fire regimes. We never take a whole clade away.

As examples we describe:

Line 174 'Fern understories remained abundant during this period and have been shown to be capable of burning intensely and carrying fires with rapid spread¹⁴.'

We go on to mention that fire adaptations appear in the Pinaceae (gymnosperms) appear 126 Ma - line 180.

Line 186 - Monocots and early eudicots appeared between 100-90 Ma and were present in conifer forests of the Northern Hemisphere⁶⁰

All of these statements highlight that we do not at any point sequentially replace any plant type but rather a new plant group is added that alters fire regimes. Therefore, we cannot add this assumption for reviewer 4 as this is not our assumption and is not at all correct. Therefore, to ask us to add this would be to add error into this manuscript.

However, I think, based on all previous comments by reviewer 4 that what is really required is for us to note in this manuscript is that our model does not capture spatially explicit changes in land plant evolution through time, nor does it recognise individual biomes. Therefore, to address this point we have added the following in the closing paragraphs of the manuscript in order to address their concerns.

“Using a simple box model, we are able to suggest that evolutionary innovations in plants over multimillion-year timescales likely have the capability to influence fire regimes and their feedback to the regulation of atmospheric oxygen. This simple approach should be built upon in further research if we are to understand the importance of long-term coupled plant based regulatory processes on our planet. Future work should seek to develop spatially explicit dynamic global vegetation models capable of operating over macro-evolutionary timescales to allow estimates of the distribution of ancient plant groups across Earth’s palaeoclimatic zones, which is a limitation of our model. Such a model would need to be coupled to a spatially explicit fire behaviour model that would allow the fire feedback to total land plant biomass to estimate the impact on P-weathering and redistribution required to influence long-term carbon burial and oxygen regulation. This would allow us to understand the role that the different biomes across climatic zones have likely played in the long-term regulation of Earth’s atmospheric composition and potentially improve their management into the future.”

The second assumption is that the early angiosperms were ‘weedy’ and highly flammable and that eventually fire resistant ‘rainforests’ developed. This assumption is debated amongst experts. Feild et al (2011) argue that the basal angiosperms (fossil and extant) have low vein density like understorey species whereas weedy species have high vein density. More research can illuminate if these assumptions are reasonable or not, it is not possible to be certain with the available evidence in hand.

Feild, T.S., Upchurch Jr., G.R., Chatelet, D.S., Brodribb, T.J., Grubbs, K.C., Samain, M.-S., Wanke, S. (2011) Fossil evidence for low gas exchange capacities for Early Cretaceous angiosperm leaves. *Paleobiology*, 37 (2), pp. 195-213. DOI: 10.1666/10015.1

I have re-read the paper again and again that reviewer 4 offers up as evidence that the earliest angiosperms were not weedy based on gas exchange but this paper does not say this. This paper suggests that high gas exchange capacities in angiosperms evolved at least 20 million years after they first appeared. Our manuscript is entirely consistent with this. We describe that high rates of gas exchange in angiosperms appeared after 100Ma see sentence starting on line 207. Prior to this we state that 'At this time [135-100 million years ago] fossil angiosperm leaves from Brazil, China, Europe, North and South America appear to have vein densities similar to Early Cretaceous ferns and gymnosperms and this likely had equal productivity potential^{57, 58}.

So we agree with Feild et al., 2011 in fact we agree and cite several of Taylor Feild's work along with other experts in this field surrounding this debate.

In response to reviewer 4's concern in this matter we have removed the word 'weedy' from the manuscript which I think is a fair statement because weediness in this context is an inference and not something that can be objectively quantified in the fossil record. Therefore, we have stuck to words that relate to measurements that can be made such as leaf mass per area or herbaceousness which can be observed from fossils. Therefore, to account for reviewer 4's comments we have edited the following sentences –

We reordered and toned down the following sentence which is the first mention of this topic

Line 123 - "It has been hypothesised that the estimated low leaf mass per area of the earliest angiosperms (135Ma) added an easily dryable and ignitable fuel load that enhanced Cretaceous fire-frequency¹⁵. From around 100 Ma angiosperms are predicted to have evolved unrivaled CO₂ uptake and transpiration capabilities as evidenced by to estimated enhanced leaf venation when compared to existing plant groups⁵⁴ ;..... "

(we moved the highlighted line upwards as it was out of time order previously and toned both lines down to be clear that they are hypotheses and estimates of specific traits).

Line 164 "We *assume that the earliest angiosperms appear ~135Ma⁵¹ and that they are of small stature and likely herbaceous ~~and weedy~~ in nature⁵⁵".

There is no literature to my knowledge that argues against the earliest angiosperms being small in stature herbaceous. In fact all basal extant angiosperms are herbaceous.

*Note we already say the word assume here also – highlighting that this is an assumption.

Line 190 - We removed weedy from in front of Ranunculaceae (as it wasn't necessary as this family is weedy).

Line 197 we swapped weedy to herbaceous.

So I feel that we have now addressed the concern here of Reviewer 4 and argue that we have provided rationale for what we implement as the fire feedback in the COPSE model and that these are not assumptions but rather a fair outlined rationale based on a reviewer of the literature in the area.

Some minor points

Line 22 the term 'tropical rainforest' here precedes the subsequent qualification of this term – edited

Line 38 some wrong with this sentence – it had an extra 'the' which we have now removed

Line 94 'fire' is used loosely clarify if this frequency, severity or both - added high fire frequency and intensity

Line 95 'suppression plant growth' add hence biomass production – added as requested

Line 136 is the claim that the angiosperms thrived in both dark and disturbed environments or dark, disturbed environments? The correct as dark and disturbed. See Feild et al., (2004) Dark and disturbed: a new image of early angiosperm ecology. *Paleobiology*, 30, 82-107.

Line 152 stressed the assumption that flammability drove angiosperm evolution – this is not certain.

Sentence updated to 'Here we update the COPSE model to qualitatively capture the effect of changes to Cretaceous regimes based on evolutionary innovations in plants, fossil evidence for increased fire activity and high oxygen driven enhancement of fire.'

Line 182 clarify you mean fire frequency – altered to fire frequency as requested.